# Inhibition of Foodborne Pathogenic Bacteria by Excreted Metabolites of *Serratia marcescens* Strains Isolated from a Dairy-Producing Environment

**DOI:** 10.3390/microorganisms11020403

**Published:** 2023-02-04

**Authors:** Bernadett Baráti-Deák, Giseli Cristina Da Costa Arruda, Judit Perjéssy, Adél Klupács, Zsolt Zalán, Csilla Mohácsi-Farkas, Ágnes Belák

**Affiliations:** 1Department of Food Microbiology, Hygiene and Safety, Institute of Food Science and Technology, Hungarian University of Agriculture and Life Sciences, Somlói út 14-16, H-1118 Budapest, Hungary; 2Department of Nutrition, Institute of Food Science and Technology, Hungarian University of Agriculture and Life Sciences, Somlói út, 14-16, H-1118 Budapest, Hungary; 3Department of Bioengineering and Fermentation Technology, Institute of Food Science and Technology, Hungarian University of Agriculture and Life Sciences, Ménesi út 45, H-1118 Budapest, Hungary

**Keywords:** food safety, natural antimicrobials, antibacterial activity, hydrolytic enzymes, prodigiosin, extracellular metabolites

## Abstract

*Serratia marcescens* strains from a dairy-producing environment were tested for their inhibitory effect on *Listeria monocytogenes*, *Salmonella* Hartford, *Yersinia enterocolitica* and *Escherichia coli*. Inhibition of foodborne pathogens was observed in the case of a non-pigmented *Serratia* strain, while the pigment-producing isolate was able to inhibit only *Y. enterocolitica*. The co-culturing study in tryptone soya broth (TSB) and milk showed that the growth of *Salmonella* was inhibited in the first 24 h, but later the pathogen could grow in the presence of the *Serratia* strain even if its cell concentration was 1000 times higher than that of *Salmonella*. However, we found that (1) concentrated cell-free supernatants had stronger inhibitory activity, which confirms the extracellular nature of the antagonistic compound(s). We proved that (2) protease and chitinase enzymes can take part in this mechanism, but they are not the main inhibitory compounds. The presence of prodigiosin was observed only in the case of the pigmented strain; thus, (3) we hypothesized that prodigiosin does not take part in the inhibition of the pathogens. However, (4) the combined effect of different extracellular metabolites might be attributed to the inhibitory property. Application of concentrated *S. marcescens* cell-free supernatant can be an effective antibacterial strategy in the food industry, mainly in the form of a bio-disinfectant on surfaces of food-processing areas.

## 1. Introduction

The number of confirmed cases for almost all zoonoses increased in the European Union in 2021 in comparison with 2020. Campylobacteriosis, salmonellosis, yersiniosis, Shiga-toxin-producing *Escherichia coli* (STEC) infections and listeriosis were the most frequently reported zoonotic diseases. Based on severity data, besides West Nile virus infections, listeriosis was the most severe disease with the highest case fatality and hospitalisation rates. The highest number of deaths was associated with listeriosis, salmonellosis, and STEC infections, with 196, 71, and 18 cases, respectively [1]; thus, control of these bacterial pathogens in foods and food-producing environments is of great importance.

Undesirable microorganisms, which may contaminate food products, food-processing facilities or manufacturing environments, pose a serious threat to consumers. Potential control strategies of these microorganisms include competitive organisms, bacteriophages, bacteriocins, siderophores, quorum sensing and different microbe- or plant-derived antimicrobials. Natural antimicrobials could inhibit microbial growth, thus contributing to the extension of food shelf-life and production of safer and nutritious foods [2,3].

*Serratia marcescens* is a Gram-negative, opportunistic nosocomial pathogen belonging to the Enterobacteriaceae family [4]. The treatment of infections caused by *S. marcescens* is a significant problem due to the intrinsic resistance of this species and its ability to acquire additional resistance to multiple groups of antimicrobial agents [5]. However, strains of *S. marcescens* originating from non-clinical environments proved to be biocontrol agents of different phytopathogenic filamentous fungi [6,7,8] and nematodes [9]. Moreover, chitinases from *S. marcescens* isolated from the gut of Chinese honey bee workers could inhibit the growth of *Varroa destructor*, an external parasitic mite [10].

Different antimicrobial strategies were identified in the case of *S. marcescens.* Among others, antibiosis by the production of pyrrolnitrin and prodigiosin [8,11], the activity of different hydrolytic enzymes [7,8,12], siderophore production [8] and type VI protein secretion system [11,13] were previously detected among *S. marcescens* strains.

Pyrrolnitrin is an antifungal antibiotic [14]; however, its antibacterial activity against streptomycetes and many phytopathogenic bacteria was proven by El-Banna and Winkelmann [15] and Chernin et al. [16]. It is produced by a large variety of bacteria such as different *Serratia* and *Pseudomonas* species, strains of *Burkholderia cepacia*, *Myxococcus fulvus*, *Corallococcus exiguous*, *Cystobacter ferrugineus* and *Enterobacter agglomerans* [17].

Prodigiosin belongs to the prodiginine family of bacterial alkaloids which are diverse sets of heterocyclic natural products [18]. It is a red pigment produced by many strains of *S. marcescens* [19], other Gram-negative bacteria (such as *Hahella chejuensis* and *Pseudoalteromonas denitrificans*) and some actinomycetes (e.g., *Streptomyces coelicolor*) [20]. Pigmented biotypes of *S. marcescens* can mostly be found in natural environments, whereas the non-pigmented biotypes are prevalent in hospitals. Moreover, pigment-negative strains are more frequent among clinical isolates than pigment-producing ones [21].

Important hydrolytic enzymes of *S. marcescens* that take part in its antimicrobial activity are chitinase, lipase and protease [8], although chitinase has a growth-inhibitory effect mainly on pathogenic and mycotoxin-producing fungi [22,23]. Besides these hydrolytic enzymes, biocontrol strains of *S. marcescens* may be able to produce siderophores. Purkayastha and co-workers [8] detected catecholate and hydroxamate types of siderophore in the case of a strain that had antagonistic effect on nine different foliar and root pathogens of tea.

Another mechanism of *S. marcescens* for growth inhibition can be the presence of a type VI protein secretion system. Murdoch and co-workers [13] observed that type VI secretion plays a crucial role in the competitiveness of *S. marcescens*, as the specialized proteinaceous machine had antibacterial killing activity against *Escherichia coli*, *Enterobacter cloacae* and *Pseudomonas fluorescens*.

In this study we aimed to characterise the antibacterial activity of two different (a pigmented and a non-pigmented) strains of *S. marcescens* isolated from a dairy-producing environment and determine their potential applicability against bacterial pathogens in the food industry.

## 2. Materials and Methods

### 2.1. Tested Bacterial Strains

*S. marcescens* CSM-RMT-1 (non-pigmented) (GenBank accession No. OQ254763) and *S. marcescens* CSM-RMT-II-1 (pigmented) (GenBank accession No. OQ254768) strains were derived from a rugged and not easily accessible surface of a diary-producing plant [24]. Pure cultures of the potential inhibitory strains were maintained on tryptone soya agar (TSA; Biokar, France) slants at 4 °C and in tryptone soya broth (TSB; Biokar, France) supplemented with 20% glycerol at −80 °C, respectively.

Inhibitory effect of the two *S. marcescens* strains were screened against four foodborne pathogenic bacteria: *Listeria monocytogenes* CCM 4699 (Czech Collection of Microorganisms, Brno, Czech Republic), *Salmonella enterica* subsp. *enterica* ser. Hartford NCAIM B.01310 (National Collection of Agricultural and Industrial Microorganisms, Budapest, Hungary), *Yersinia enterocolitica* HNCMB 98002 (Hungarian National Collection of Medical Bacteria, Budapest, Hungary) and *Escherichia coli* NCAIM B.01909 (National Collection of Agricultural and Industrial Microorganisms, Budapest, Hungary). Each of them was cultured on TSA at 37 °C for 24 h, except *Y. enterocolitica,* which was incubated at 25 °C.

In the co-culturing study the growth of a food-derived *Salmonella enterica* subsp. *enterica* strain was tested. This bacterium was previously isolated from egg and identified by Sanger sequencing of the 16S rRNA encoding gene and MALDI-TOF MS analyses (data are not shown). The GenBank accession number of the strain is OQ254770. The pathogen was cultured on TSA at 37 °C for 24 h, while its maintenance was performed on TSA slants at 4 °C and in TSB supplemented with 20% glycerol at −80 °C.

### 2.2. Screening for Antibacterial Activity of S. marcescens Strains and Testing the Inhibitory Effect of Their Cell-Free Supernatants

*S. marcescens* CSM-RMT-1 and *S. marcescens* CSM-RMT-II-1 were screened for their antibacterial activity against the four foodborne pathogens using the agar-spot method, while the production and effect of extracellular inhibitory substances were examined using cell-free supernatants of the test strains by micro-culturing.

In the case of the agar spot method, 2.5 McFarland (approx. 10^8^–10^9^ cells mL^–1^) suspensions of the pathogenic bacteria were prepared in sterile distilled water using overnight cultures. From the ten-fold dilutions, aliquots of 0.1 mL (with a final concentration of 10^6^ cells mL^–1^) were massively inoculated onto TSA plates, and after drying, 10 μL of cell suspension made of *S. marcescens* isolates (containing approx. 10^6^ cells) was dropped onto the agar surface. The plates were incubated at 5, 10, 15, 20, 25, 30, 37 and 42 °C for 6 days. Growth inhibition of the pathogens was observed by checking the clearing zones around the colonies of *S. marcescens* strains after one, two, three and six days of incubation. The experiment was performed in triplicate.

One-, three- and six-day-old cell-free supernatants of *S. marcescens* CSM-RMT-1 were generated from cells cultured in TSB at 25 °C. After centrifugation at 14,000× r.p.m. for 15 min the supernatants were removed and filtered through a 0.2 μm pore-size membrane filters (FilterBio PES Syringe Filter, Lab-Ex Ltd., Budapest, Hungary).

The ten-times-concentrated samples were prepared from cell-free supernatants by freezing the liquid at −80 °C then lyophilized using a Scanvac CoolSafe^TM^ freeze dryer (LaboGene, Lillerød, Denmark). The freeze-dried samples were soaked in one tenth amount of TSB; therefore, the inhibition assays could be repeated with ten times more concentrated supernatants using the micro-culturing method.

Multiskan Ascent (Thermo Fisher Scientific, Waltham, MA, USA) was used for micro-culturing to determine the production of extracellular inhibitory substances in the case of *S. marcescens* strains. The wells of the microplates were filled with 300 μL of liquid consisting of 75 μL of four-fold strength TSB, 75 μL of cell suspension of the appropriate pathogen (approx. 10^6^ CFU mL^–1^), and 150 μL cell-free supernatant. Incubation of the microplates was performed at 25 °C, and the absorbance values were recorded automatically every 30 min during 24 h of cultivation at 595 nm. Growth curves were generated from the absorbance values versus time data using the averages of triplicates. In each experiment, the growth of the pathogenic bacterium was tested alone (without the presence of cell-free supernatant), which represented the control sample.

### 2.3. Cellophane Test

Based on the results of the supernatants’ inhibitory effect, a cellophane test was carried out with the *S. marcescens* CSM-RMT-1 strain. The analysis was conducted as described previously in Baráti-Deák et al. [24].

### 2.4. Detection of Proteolytic and Chitinase Activities of S. marcescens Strains

Proteolytic activity of the two *S. marcescens* strains was tested using skim milk agar (SMA) made from the following components (each purchased from Merck Life Science Ltd., Budapest, Hungary): yeast extract 2.5 g/L, peptone 1 g/L, glucose 5 g/L, milk powder 10 g/L and agar-agar 15 g/L. A cell suspension with a 0.5 McFarland standard turbidity was prepared in distilled water. Ten μL of the suspension was dropped onto the surface of SMA plates, which were incubated at 20, 25 and 30 °C. The diameters of the appearing clearing zones were measured after one, two and five days of incubation. The test was performed in triplicate. Statistical analyses were carried out with IBM SPSS Statistics 22 software (IBM Corporation, Armonk, NY,Endicott, NY, USA).

For the detection of chitinase production and activity a method described by Leisner et al. [25] was applied. Cell suspensions of the tested *S. marcescens* strains were dropped onto basic chitin medium (BMC) (triptone 10 g/L, yeast extract 5 g/L, NaCl 10 g/L, agar 15 g/L (each purchased from Merck Life Science Ltd., Hungary) and α-kitin (Sigma-Aldrich, Budapest, Hungary) 2.5 g/L), and the presence of clearing zones was checked after three days of incubation at 30 °C. Later, the method was slightly modified, as colloidal chitin made from shrimp-shell powder and shrimp-shell coarse flakes was used as substrate according to Liu and co-workers [26]. Briefly, 4 g of powdered chitin or flakes was suspended in 40 mL of 37% HCl and mixed for 50 min. Then, 1 L of ice-cold water was added dropwise. After centrifugation at 8000× *g* for 20 min, the pellet was collected and washed with distilled water until the pH of the washing water reached 5.0. In this way, prepared and then dried colloidal chitin was added as substrate to the BMC medium instead of α-chitin in 1 g/L concentration.

### 2.5. Detection of Prodigiosin-Producing Ability of S. marcescens Strains

The colour test for prodigiosin production was conducted according to Bharmal et al. [27] with minor modifications. *S. marcescens* CSM-RMT-1 and CSM-RMT-II-1 were cultured on Nutrient agar plates (Merck Life Science Ltd., Budapest, Hungary) and incubated overnight at the following temperatures: 20, 25, 30, 37 and 40 °C. The prodigiosin extraction was performed using 3–4 loopfuls of the overnight cultures mixed with 1 mL 96% ethanol (Reanal, Hungary) in an Eppendorf tube. It was set at room temperature for 24 h. After centrifugation at 10,000× *g* for 15 min either 10 μL of 37% HCl or 10 μL of NH**_3_** was added to the pellet in a test tube. The appearing red (acidic environment) or yellow (alkaline environment) colour change was detected in the case of the presence of prodigiosin [27].

### 2.6. PCR Analysis of Prodigiosin- and Chitinase-Encoding Genes

For testing the presence of the *pig* gene cluster, which is responsible for prodigiosin production, PCR amplifications of *cueR* and *copA* genes were performed as described by Harris et al. [28]. The *cueR* is responsible for copper efflux regulation, while *copA* is a copper-transporting P-type ATPase, and these two genes flank the pigment cluster [28]. The chitinase encoding *chiA* gene of *S. marcescens* strains was amplified with the method of Ramaiah et al. [29]. The applied primers are listed in Table 1.

### 2.7. Co-Culturing of S. marcescens CSM-RMT-1 with a Food-Originated Salmonella enterica Subsp. enterica in Culture Broth

Co-culturing study of a food-derived *Sa. enterica* subsp. *enterica* strain (GenBank accession No. OQ254770) was carried out using *S. marcescens* CSM-RMT-1 as the inhibitory strain. As a non-food model environment, TSB was used as culturing broth under static conditions (without shaking). Flasks containing TSB were inoculated with cells of *Sa. enterica* and *S. marcences* CSM-RMT-1 in volume ratios of 1:1 and 1:1000, respectively. Growth of *Sa. enterica* and *S. marcences* CSM-RMT-1 were also tested in culture broth separately; these samples represented the controls. The flasks were incubated for six days at 25 °C. Samples were taken after one, two, three and six days of incubation, and cell counts were determined with the spread plate method using TSA and *Salmonella* selective Harlequin^TM^ agar (Lab M Limited, Lancashire, UK) plates. The *Salmonella* and *Serratia* colonies were counted together on TSA plates, while on Harlequin plates, only the green *Salmonella* colonies were enumerated. Therefore, the colony number of *Serratia* was determined from the TSA plate count minus the *Salmonella* counted on Harlequin agar. Each analysis was conducted in triplicate, and statistical analyses were carried out using Microsoft Excel software’s functions.

The co-culturing study was also performed by replacing TSB with UHT milk (containing 2.8% fat) as a food matrix. All parameters and settings were the same as in the previous experiment.

### 2.8. Statistical Analysis

Data were tested for normality using the Kolmogorov–Smirnov and Shapiro–Wilk tests, while the homogeneity of variance was tested using the Levene test. According to the results of these tests, Games–Howell or Tukey HSD nonparametric post hoc tests were used to determine significant differences at a significance level of 0.05. To indicate the significant differences in pairwise comparisons, upper-case letters were used on the columns of graphs according to the compact letters display (CLD). For the statistical analyses, IBM SPSS Statistics software (version 22) (IBM, Armonk, New York, NY USA) was used.

## 3. Results

### 3.1. Inhibitory Activity of the Tested S. marcescens Strains on Different Foodborne Pathogenic Bacteria and the Effect of Cell-Free Supernatants

*S. marcescens* CSM-RMT-1 had a growth-inhibitory effect on all four pathogenic bacteria (Table 2) as determined by the agar spot method; however, *Sa*. Hartford was only partially inhibited. Its impact was the strongest against *Y. enterocolitica* as total inhibition could be observed during the six-day-long incubation. In the case of *L. monocytogenes* and *E. coli,* complete growth decline was only detected on the first and second days of incubation; later, the pathogens could overgrow *S. marcescens* CSM-RMT-1, and there was only a partial inhibitory effect at the end of the incubation period (Figure 1).

The strain *S. marcescens* CSM-RMT-II-1 was effective only against *Y. enterocolitica* at 30 °C, as it could totally inhibit its growth during the six days of incubation. At the same time, this strain proved to be ineffective against the other three bacterial pathogens (Figure 1).

The optimal temperature range for inhibition was tested at eight different temperatures (from 5 to 42 °C) and proved to be between 15 and 30 °C in the case of *S. marcescens* strains (Table 3).

Effects of cell-free supernatants were examined only on those pathogenic bacteria that were inhibited by the *S. marcescens* strains in the agar spot method. Examining the effect of extracellular metabolites, we found that the cell-free supernatant of S. *marcescens* CSM-RMT-II-1 inhibited the growth of *Y. enterocolitica*, similarly to what was observed in the contact inhibition test. In the case of strain CSM-RMT-1 (which had a negative impact on all pathogens in the agar spot method), the inhibitory effect on *Y. enterocolitica* was not as strong as it could be expected on the basis of the results for contact inhibition. Moreover, its supernatants were not in the least effective against *L. monocytogenes*, *E. coli* and *Sa.* Hartford during the tested time period (Figure 2A–D).

By applying the cellophane test, we wanted to determine whether *S. marcescens* CSM-RMT-1 could inhibit the pathogens without the presence of its cells using a solid medium. Thus, in the framework of this study, the effect of diffusible extracellular metabolites was tested on the agar plates. The results showed that the inhibitory effect of *S. marcescens* CSM-RMT-1 was not weakened in this test compared to the results of contact inhibition, which confirms the assumption that the produced antagonistic extracellular compounds were able to diffuse into the agar through the cellophane layer and affect the pathogens’ growth in concentrated form.

Based on these observations the cell-free supernatants were lyophilised and concentrated, and the effects of concentrates (ten times stronger supernatants) were tested again. As can be seen in Figure 1A–D, the concentrated supernatants of the *S. marcescens* CSM-RMT-1 strain were more effective against all four tested pathogens than the non-lyophilised ones. Total inhibition of *L. monocytogenes*, *Y. enterocolitica* and *E. coli* could be detected; however, in case of *Sa*. Hartford only the lag phase was prolonged and a slight increase in cell number was observed after 10 h of incubation.

### 3.2. Protease and Chitinase Enzyme Activities of the S. marcescens Strains, and Their Prodigiosin Production

From the results of the proteolytic tests, it can be seen that protease activities became stronger in both *S. marcescens* strains by the progress of the incubation time (Figure 3). An approximately three- to five-times higher increase in the size of clearing zones could be observed between the results of 24 and 120 h incubations, respectively. This was confirmed with statistical analyses, which showed significant differences (*p* < 0.05) between the proteolytic activities of the strains at different time points of incubation (Table 4). The only exception was seen at 20 °C, where no significant difference was found between the 24- and 48-h samples.

Regarding the incubation temperatures, the best results were seen at 30 °C; however, all tested temperatures proved to be sufficient for protease enzyme production, since it did not have any significant effect (*p* > 0.05) on the proteolytic activity. Moreover, there were no significant differences (*p* > 0.05) between the proteolytic activities of the two *S. marcescens* strains (Figure 3, Table 4). Nonetheless, their inhibitory patterns were varied.

Chitinase activity of the two *S. marcescens* strains was detected by applying colloidal chitin. Based on the clearing zones around the colonies, both *Serratia* were capable of producing chitinase enzyme. According to the sizes of the clearing zones, the chitinase activity of strain CSM-RMT-1 (3.2 ± 0.5 mm) was greater than that of strain CSM-RMT-II-1 (1.1 ± 0.4 mm).

Prodigiosin production was tested using the extracts generated from overnight cultures of the two *S. marcescens* strains. The presence of prodigiosin was observed in the case of strain CSM-RMT-II-1 between 20 and 30 °C using acidic as well as alkaline extraction. However, strain CSM-RMT-1 did not show any colour changes when applying either acidic or alkalic conditions. These results are in accordance with our culturing observations, as strain CSM-RMT-1 has white colonies, while strain CSM-RMT-II-1 forms red colonies when cultured on TSA and Nutrient agar plates, respectively. The higher temperatures (37 and 42 °C) were not as appropriate for prodigiosin production as the lower ones (between 20 and 30 °C) since there were only slight changes in the colour of the test tubes under both alkaline and acidic conditions. However, at lower temperatures strong colour changes were observed.

### 3.3. Presence of Prodigiosin- and Chitinase-Encoding Genes of S. marcescens Strains

After the colour tests for prodigiosin production, PCR detection of *cueR*-*pigA* and *pigN-copA* regions were performed in the case of both *S. marcescens* strains. There were no PCR products observed for CSM-RMT-1 for either the *cueR-pigA* or the *pigN-copA* regions on the agarose gels after electrophoresis.

In the case of the CSM-RMT-II-1 strain, primers specific for the *pigN-copA* segment could generate amplicons; however, their binding was not specific as the size of the PCR product was bigger (around 500 bp) than the expected 244 bp. Further optimisation regarding the reaction compounds and the annealing temperature was conducted, but the amplicon was still larger than expected. The presence of a DNA segment (with approx. 480 bp) produced by the cueR and PE1 primer pair (specific for *cueR*-*pigA* region) could be observed, which means that this strain harbours the *cueR* and *pigA* genes. These results are in agreement with the observations of the prodigiosin-production and culturing studies.

In the case of *chiA*-specific PCR, both strains generated a 225 bp long amplicon, which refers to the fact that the two *S. marcescens* strains harbour the gene coding for chitinase enzyme.

### 3.4. Results of Co-Culturing of Prodigiosin-Negative S. marcescens CSM-RMT-1 with A Food-Derived Salmonella enteritica Subsp. enteritica Strain

As the non-pigmented CSM-RMT-1 strain had a good antagonistic effect against the four tested pathogens, a *Sa. enterica* strain isolated from egg powder was chosen to observe the antagonistic activity of *S. marcescens* CSM-RMT-1 cells in TSB and milk by co-culturing.

*S. marcescens* CSM-RMT-1 had a negative effect on the growth of the pathogen in TSB, which was more remarkable in the case of the 1:1000 volume ratio; however, in both cases the inhibition was significant (α = 0.05) (Table 5). As shown in Figure 4A, the antagonistic effect caused a nearly two-log decline in the number of *Salmonella* after the first day of incubation. However, it can also be seen that at the sixth day there was no significant difference between the numbers of viable *Salmonella* cells in the case of the diverse volume ratios (Figure 4A).

Using milk as a food matrix, *S. marcescens* CSM-RMT-1 showed a better antagonistic effect in both of the used volume ratios (1:1 and 1:1000), as the numbers of *Salmonella* were at least one log less than in case of the experiment with culture broth (Figure 4B). During the incubation period the pathogen could slowly grow again and on the last day of the experiment the difference between the numbers of *Salmonella* in each mixture was less than one log; however, it was still significant (α = 0.05) compared to the control (Table 5).

## 4. Discussion

*S. marcescens* is a widely distributed saprophytic bacterium. It is a normal commensal of the alimentary tract and can be found in food as well, particularly in starchy variants that provide an excellent growth environment for it [5,30]. In food-processing environments, numerous Enterobacteriaceae genera are present, and out of these bacteria *Serratia* species may be detected from meat, poultry, salmon, milk and ready-to-eat (RTE) food-processing plants [31]. Our potential biocontrol *S. marcescens* strains were isolated from a dairy-food-producing environment. Amorim et al. [32] isolated *S. marcescens* pigment-producing strains from dairy products as well; however, these isolates were multidrug resistant strains, and the study focused mainly on the health problems associated with them and not on their possible application as biocontrol strains in the food industry. In another article, isolation of *S. marcescens* strains from stainless steel surfaces in pre- and post-pasteurization pipelines of a milk-processing plant was reported [33]. In addition to numerous literature sources, it is also confirmed by the aforementioned references and our results that the appearance of *S. marcescens* strains in food-processing areas and in milk-processing plants is widespread, and both pigment-producing and non-pigmented strains can be found in these places.

Many isolates of environmental *Serratia* species can colonize a wide range of ecological niches, adapt to harsh environmental circumstances and compete with other bacteria by producing siderophores and compounds inhibiting the growth of other bacteria. These metabolites can be extracellular products including chitinases, proteases, lipases, nucleases, bacteriocins, surfactants and wetting agents [28,31]. In our study we demonstrated that the tested *S. marcescens* strains were able to inhibit four food-borne pathogenic bacteria, of which three were Gram-negative and one was Gram-positive. Presumably, prodigiosin does not take part in the inhibition of pathogens as *S. marcescens* CSM-RMT-1 is not able to produce this pigment; however, it could negatively influence the growth of all tested pathogenic bacteria. Prodigiosin, a bioactive secondary metabolite with antibacterial, antifungal and antiprotozoal activity [34], had antibacterial activity against pathogenic Gram-positive bacteria (*Staphylococcus aureus*, *Enterococcus faecalis* and *Streptococcus pyogenes*) but was ineffective against Gram-negative pathogens [35]. Interestingly, our prodigiosin-producing CSM-RMT-II-1 *S. marcescens* strain was able to inhibit only *Y. enterocolitica*, a Gram-negative food-borne pathogen.

The *pig* gene cluster responsible for the red pigment production can be flanked by the genes *cueR* and *copA* as Harris and co-workers [28] found in case of *S. marcescens* ATCC 274. They also observed that this configuration is demonstrated in several *S. marcescens* strains, whilst these genes are contiguous in strains lacking the *pig* cluster. In our study PCR detection of these flanking genes was performed using specific primer pairs published by Harris et al. [28]. There were no PCR products in the case of the non-pigmented CSM-RMT-1 *S. marcescens* strain using either the cueR-PE1 or the ab77-PE2 primers. This means that this strain presumably does not contain the genes of the *pig* cluster in its genome or have cryptic *pig* genes. These PCR results together with those of the previous colour extraction ones can suggest that *S. marcescens* CSM-RMT-1 is unable to produce prodigiosin, and therefore its inhibitory effect cannot be caused by this metabolite.

In the case of CSM-RMT-II-1 the ab77 and PE2 primers could attach to the *pigN*-*copA* region, but the binding was not specific, as an amplicon with 500 bp size was generated. The expected amplicon is around 244 bp. At the same time, a DNA segment of approximately 480 bp could be observed in the case of *cueR*-*pigA*-region amplification, and although its size is smaller than the expected one, Venil et al. [36] observed the presence of a 488 bp long PCR product generated by the cueR-PE1 primer pair in the case of *S. marcescens* SB08 typical to this region. Thus, our result could refer to the presence of the *pig* gene cluster in the genome of *S. marcescens* CSM-RMT-II-1. This is supported by the observation of Harris et al. [28], as they noticed that if *cueR*–*copA* genes were adjacent a small PCR product with approx. 173 bp would appear on the agarose gel instead of the 505 and 244 bp long amplicons.

Significant differences were not found between the protease activities of our two *S. marcescens* strains; however, their inhibitory patterns were variant. This may refer to the fact that proteases can take part in the inhibition of pathogens, but they are not the main inhibitory compounds of our antagonistic *S. marcescens* strains. Moreover, protease production of *Serratia* species could contribute to the spoilage of milk [31]; thus, a direct application of the cell-free supernatants gained from our biocontrol strains would negatively influence the quality of milk products.

The chitinase production of *Serratia* species is a well-known characteristic. *S. marcescens* is one of the most efficient chitin degraders amongst bacteria [37,38,39]; however, this ability is strain-dependent [23]. The presence of the chitinase-coding gene and chitinase activity of the isolated CSM-RMT-1 and CSM-RMT-II-1 strains was proved. The studied strains contain the *chiA* gene, which widely occurs among *S. marcescens* strains, such as *S. marcescens* TRL [40], *S. marcescens* 2170 [41] and *S. marcescens* BJL200 [42]. The activity of chitinase in the case of our strains was obviously visible both on chitin powder and colloidal-chitin-containing agar plates. However, the production of chitinases is influenced by several environmental parameters, the culture conditions and the components of growth medium [23,38].

The co-culturing examinations showed that the non-pigmented CSM-RMT-1 *S. marcescens* strain was able to negatively affect the growth of a food-derived *Sa. enterica* isolate both in culture broth and in milk when present at a 1000-times-higher concentration compared to the pathogen; however, cell numbers of the tested bacteria increased significantly during the first 24 h of incubation, which could lead to the spoilage of the food matrix (caused by *Serratia marcescens*). However, milk is a good medium for bacterial growth and has natural antimicrobial agents such as lactoferrin, lactoperoxidase, lysozyme and some imunnoglobulines [43], which could contribute to the antagonistic effect of the tested *Serratia* strain and therefore cause a slightly better inhibition compared to the result of TSB.Thus, direct application of the antagonistic cells is not advisable for biocontrol purposes, but the usage of their concentrated and purified cell-free supernatants in the food industry can be an appropriate alternative for the inhibition of bacterial pathogens. At the same time investigations concerning the safe utilization of such products are crucial and indispensable.

*S. marcescens* biocontrol strains were isolated and characterized earlier, but the reported bacteria can be used for field application in order to minimize the use of chemical fungicides and insecticides for disease control [8,44,45,46,47]. The isolated and investigated *S. marcescens* strains investigated in this study have potential as effective and persistent biological control agents for food industrial application.

## 5. Conclusions

Molecules responsible for the inhibitory activity of the studied *S. marcescens* strains are probably extracellular metabolites, as in the cellophane test, we proved that the presence of cells of the antagonistic strains is not necessary, and the diffusible compounds produced by the strains have the same effect as the cells themselves. Prodigiosin has no relevant role in the inhibition, as the *S. marcescens* CSM-RMT-1 strain is not able to produce this pigment, yet has good inhibitory activity against the tested pathogens. Nevertheless, different hydrolytic enzymes that are able to break down the essential macromolecules of the target cells are most likely the inhibitory substances of our *S. marcescens* strains. The antagonistic *S. marcescens* contains the *chiA* gene and produces chitinase, which could also have antifungal and antibacterial activity. Based on our results and observations, these bacteria—mainly the CSM-RMT-1 strain—or rather their concentrated cell-free supernatants can be used as antagonists of different pathogenic bacteria in the food industry, mainly in the form of bio-disinfectants on surfaces of food-processing areas. However, further analyses should be carried out to verify whether diffusible proteins or any other extracellular compounds might (or might not) affect the composition of foods or food raw materials.

## Figures and Tables

**Figure 1 microorganisms-11-00403-f001:**
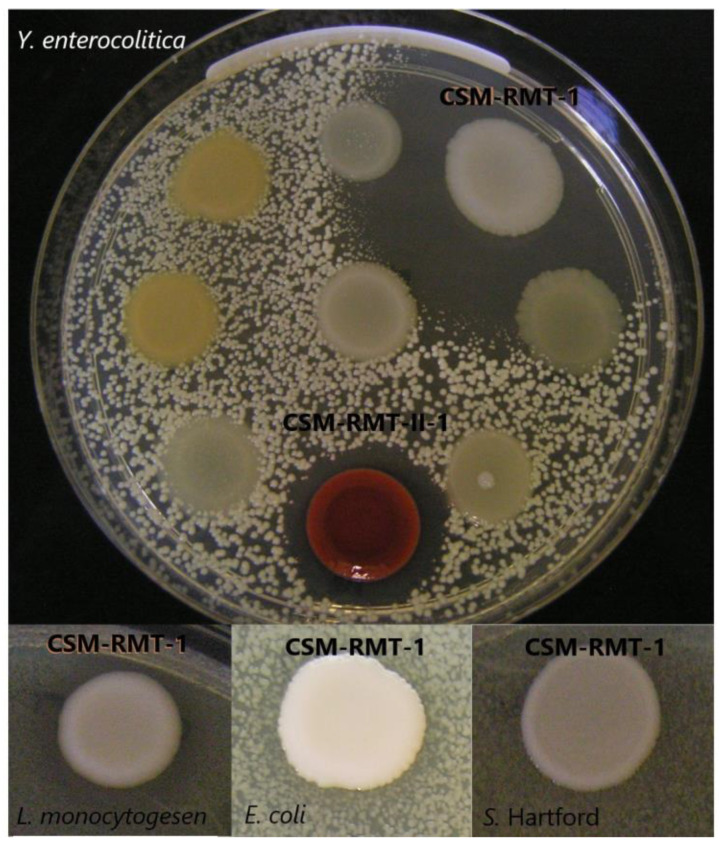
Effect of *S. marcescens* CSM-RMT-1 and CSM-RMT-II-1 strains on growth of *Y. enterocolitica, L. monocytogenes* and *E. coli* after 6 days of incubation at 30 °C detected with the agar spot method. In the case of *Salmonella* Hartford the zone was detected at 10 °C on day 3.

**Figure 2 microorganisms-11-00403-f002:**
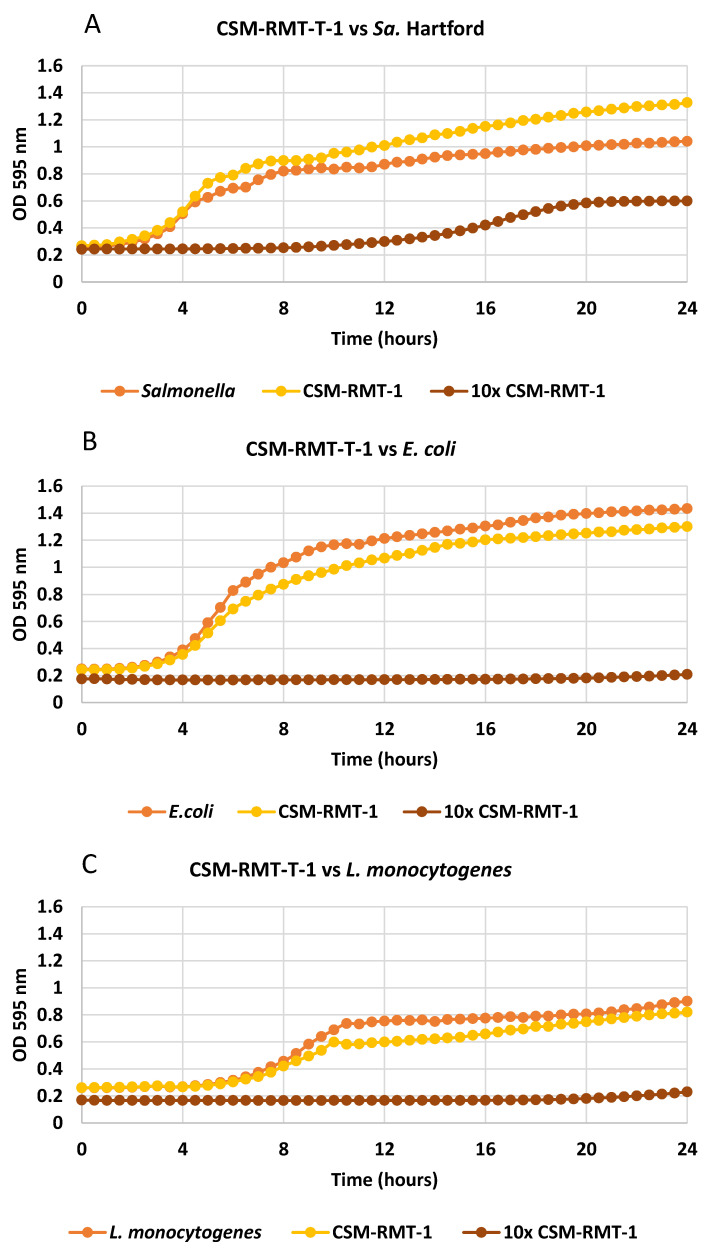
(**A**–**D**). Effect of non-lyophilised and concentrated (10×) cell-free supernatants of *S. marcescens* CSM-RMT-1 on growth of the tested foodborne pathogenic bacteria ((**A**): *Salmonella* Hartford, (**B**): *E. coli*, (**C**): *L. monocytogenes*, (**D**): *Y. enterocolitica*).

**Figure 3 microorganisms-11-00403-f003:**
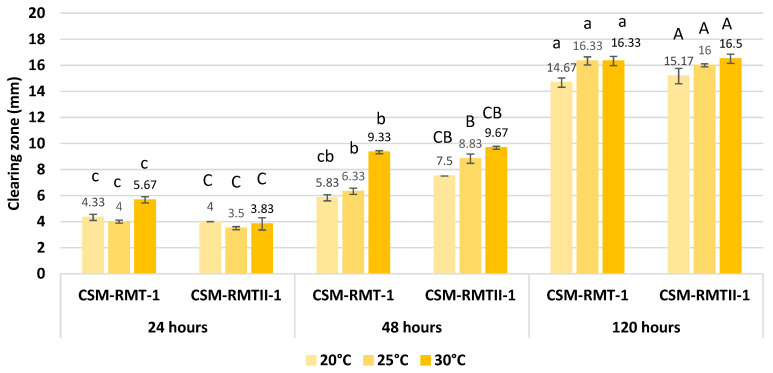
Proteolytic activities of *S. marcescens* CSM-RMT-1 and CSM-RMT-II-1 strains after 24, 48 and 120 h of incubation at three different (20, 25 and 30 °C) temperatures. Sizes of clearing zones are mean values of three parallel measurements. The bars indicate the standard deviation of three replicates of the experiment. Different upper-case letters between the incubation times indicate a significant difference between the clearing zone within the same strain according to Games–Howell post hoc tests (α = 0.05).

**Figure 4 microorganisms-11-00403-f004:**
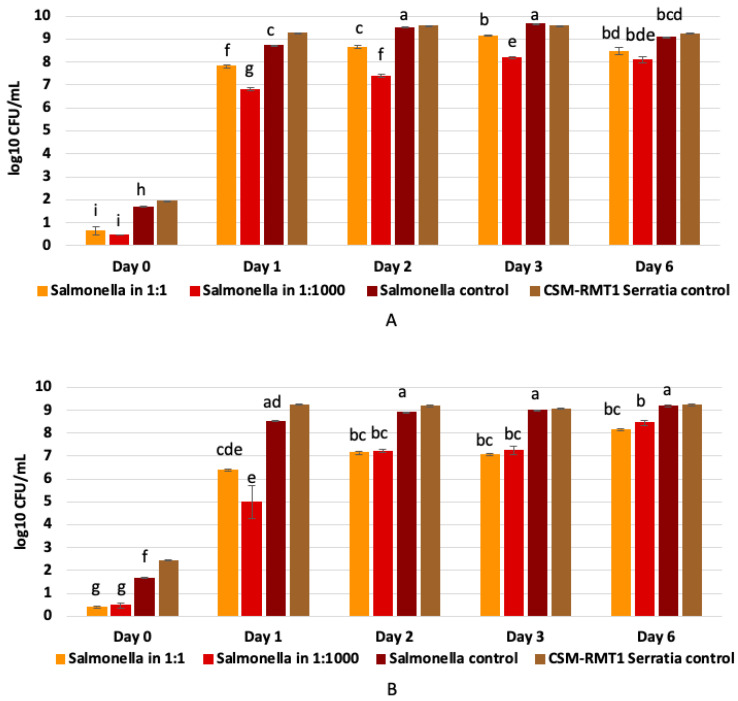
Co-culturing results of prodigiosin-negative *S. marcescens* CSM-RMT-1 and *Sa. enterica* in (**A**) TS broth and (**B**) 2.8% fat-containing UHT milk after 0, 1, 2, 3 and 6 days of incubation. Different upper-case letters by CLD indicate a significant difference between the cell numbers according to Tukey’s HSD post hoc tests (α = 0.05).

**Table 1 microorganisms-11-00403-t001:** Primer pairs used for PCR amplification of genes encoding for prodigiosin (*ceuR* and *copA*) and chitinase (*chiA*) of *S. marcescens* strains.

Target Region or Gene	Primer Name	Sequence of the Primer (5′–3′)	Size of the PCR Product (bp)	References
*cueR*-*pigA*	cueR	TCGTAAAAACGAATCGTC	505	[28]
	PE1	GCAAAACTCTGAGCGGATTCGC		
*pigN-copA*	ab77	GAAACACTTAACCTGACG	244	
	PE2	CGCAGTTCATGCAGGACAGC		
*chiA*	chiFEMSF	GATATCGACTGGGAGTTCCC	225	[29]
	chiFEMSR	CATAGAAGTCGTAGGTCATC		

**Table 2 microorganisms-11-00403-t002:** Inhibitory effect of *S. marcescens* CSM-RMT-1 and CSM-RMT-II-1 strains on the tested foodborne pathogenic bacteria examined with the agar spot method after one, two, three and six days of incubation (based on reference [24]).

Tested Foodborne Bacterial Pathogens	Potential Inhibitory *S. marcescens* Strains	
	CSM-RMT-1	CSM-RMT-II-1
*L. monocytogenes* CCM 4699	+	−
*Sa.* Hartford NCAIM B.01310	(+)	−
*Y. enterocolitica* HNCMB 98002	+	+
*E. coli* NCAIM B.01909	+	−

(+): partial inhibition, −: no inhibition, +: total inhibition.

**Table 3 microorganisms-11-00403-t003:** Inhibitory effect of *Serratia marcescens* CSM-RMT-1 at different incubation temperatures on growth of the pathogenic bacteria.

Days of Incubation	Tested Temperatures							
	5 °C	10 °C	15 °C	20 °C	25 °C	30 °C	37 °C	42 °C
	**Inhibition of *Sa.* Hartford by *S. marcescens* CSM-RMT-1**							
1	−	−	−	−	−	−	−	−
2	−	−	−	−	−	−	−	−
3	−	(+)	−	−	−	−	−	−
6	−	−	−	−	−	−	−	−
	**Inhibition of *E. coli* by *S. marcescens* CSM−RMT−1**							
1	−	−	−	−	−	+	−	−
2	−	−	−	−	−	(+)	−	−
3	−	−	−	−	−	(+)	−	−
6	−	−	−	−	−	(+)	−	−
	**Inhibition of *L. monocytogenes* by *S. marcescens* CSM-RMT-1**							
1	−	−	−	(+)	(+)	−	−	−
2	−	−	+	(+)	(+)	(+)	−	−
3	−	(+)	+	(+)	(+)	(+)	−	−
6	−	(+)	+	(+)	−	(+)	−	−
	**Inhibition of *Y. enterocolitica* by *S. marcescens* CSM−RMT−1**							
1	−	−	−	−	−	+	−	−
2	−	−	−	+	+	+	−	−
3	−	−	−	+	+	+	−	−
6	−	−	+	+	+	+	−	−

−: no inhibition, (+): partial inhibition, +: total inhibition

**Table 4 microorganisms-11-00403-t004:** Statistical results of protease activity in the case of *S. marcescens* CSM-RMT-1 and *S. marcescens* CSM-RMT-II-1.

Time	Temp.	Mean	Std. Deviation	24 h			48 h			120 h		
				20 °C	25 °C	30 °C	20 °C	25 °C	30 °C	20 °C	25 °C	30 °C
*S. marcescens* CSM-RMT-1												
24 h	20 °C	4.333	0.5774		0.951	0.319	0.173	0.086	0.004	0.000	0.004	0.001
	25 °C	4.000	0.0000			0.168	0.038	0.024	0.018	0.005	0.014	0.001
	30 °C	5.667	0.5774				1.000	0.699	0.013	0.000	0.006	0.001
48 h	20 °C	5.833	0.2887					0.553	0.019	0.001	0.013	0.000
	25 °C	6.333	0.2887						0.029	0.002	0.015	0.000
	30 °C	9.333	0.5774							0.003	0.019	0.003
120 h	20 °C	14.667	0.5774								0.533	0.135
	25°C	16.333	1.1547									1.000
	30 °C	16.333	0.2887									
*S. marcescens* CSM-RMT-II-1												
24 h	20 °C	4.000	0.0000		0.951	1.000	0.091	0.038	0.004	0.001	0.008	0.008
	25 °C	3.500	0.8660			1.000	0.042	0.013	0.019	0.004	0.001	0.000
	30 °C	3.833	1.4434				0.183	0.084	0.083	0.021	0.005	0.005
48 h	20 °C	7.500	0.8660					0.604	0.198	0.011	0.003	0.002
	25 °C	8.833	0.7638						0.710	0.012	0.004	0.003
	30 °C	9.667	0.2887							0.000	0.018	0.015
120 h	20 °C	15.167	0.2887								0.780	0.463
	25 °C	16.000	0.8660									0.995
	30 °C	16.500	0.8660									

Coloured cells indicate significant differences between the clearing zones according to Games–Howell post hoc tests (α = 0.05).

**Table 5 microorganisms-11-00403-t005:** Results of statistical analysis of co-culturing in the case of *Sa. enterica* and *S. marcescens* CSM-RMT-1 in TSB and milk. Significance is expressed by *p*-value.

Sample	Sampling Time	Mean	Std. Deviation	*Salmonella* Control					*Salmonella* 1:1					*Salmonella* 1:1000				
				Day 0	Day 1	Day 2	Day 3	Day 6	Day 0	Day 1	Day 2	Day 3	Day 6	Day 0	Day 1	Day 2	Day 3	Day 6
Co-culturing in TSB																		
*Salmonella* control	Day 0	1.666	0.0205		0.000	0.000	0.000	0.000	0.000	0.000	0.000	0.000	0.000	0.000	0.000	0.000	0.000	0.000
	Day 1	8.528	0.0254			0.000	0.000	0.214	0.000	0.000	1.000	0.041	0.039	0.000	0.000	0.000	0.000	0.048
	Day 2	8.897	0.0311				0.968	0.460	0.000	0.000	0.000	0.020	0.000	0.000	0.000	0.000	0.000	0.000
	Day 3	8.991	0.0188					0.006	0.000	0.000	0.000	0.017	0.000	0.000	0.000	0.000	0.000	0.000
	Day 6	9.185	0.0738						0.000	0.000	0.080	1.000	0.057	0.000	0.000	0.000	0.000	0.051
*Salmonella* 1:1	Day 0	0.391	0.0086							0.000	0.000	0.000	0.000	0.947	0.000	0.000	0.000	0.000
	Day 1	6.389	0.1245								0.000	0.000	0.005	0.000	0.000	0.101	0.018	0.000
	Day 2	7.135	0.1336									0.028	0.037	0.000	0.000	0.000	0.042	0.020
	Day 3	7.056	0.0805										0.110	0.000	0.000	0.000	0.000	0.120
	Day 6	8.147	0.1022											0.000	0.000	0.000	0.000	0.929
*Salmonella* 1:1000	Day 0	0.473	0.0016												0.000	0.000	0.000	0.000
	Day 1	5.000	1.4142													0.007	0.000	0.000
	Day 2	7.223	0.1095														0.000	0.000
	Day 3	7.238	0.3373															0.091
	Day 6	8.451	0.2128															
Co-culturing in Milk																		
*Salmonella* control	Day 0	1.693	0.0296		0.000	0.000	0.000	0.000	0.009	0.000	0.000	0.000	0.000	0.008	0.000	0.000	0.000	0.000
	Day 1	8.771	0.0059			0.999	0.993	0.902	0.000	0.003	0.003	0.024	0.049	0.000	0.000	0.039	0.041	0.049
	Day 2	9.501	0.0386				1.000	1.000	0.000	0.001	0.020	0.014	0.480	0.000	0.000	0.030	0.032	0.048
	Day 3	9.667	0.0198					1.000	0.000	0.000	0.013	0.009	0.049	0.000	0.000	0.020	0.021	0.049
	Day 6	9.069	0.0574						0.000	0.000	0.005	0.004	0.041	0.000	0.000	0.008	0.009	0.039
*Salmonella* 1:1	Day 0	0.652	0.0287							0.000	0.000	0.000	0.000	0.904	0.000	0.000	0.000	0.000
	Day 1	7.801	0.1443								0.817	0.901	0.045	0.000	0.341	0.699	0.677	0.015
	Day 2	8.642	0.1388									1.000	0.441	0.000	0.004	0.695	1.000	0.145
	Day 3	9.138	0.0490										0.341	0.000	0.005	1.000	1.000	0.104
	Day 6	8.404	0.3234											0.000	0.000	0.566	0.588	0.906
*Salmonella* 1:1000	Day 0	0.474	0.0248												0.000	0.000	0.000	0.000
	Day 1	6.801	0.1443													0.002	0.002	0.000
	Day 2	7.389	0.1245														1.000	0.206
	Day 3	8.172	0.0823															0.218
	Day 6	8.590	0.0157															

Coloured cells indicate significant differences between the *Salmonella* cell number according to Tukey’s HSD post hoc tests (α = 0.05).

## Data Availability

The datasets generated during and/or analyzed during the current study are available from the corresponding author on request.

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
