# Peer review of "Inhibition of Foodborne Pathogenic Bacteria by Excreted Metabolites of Serratia marcescens Strains Isolated from a Dairy-Producing Environment"

_microorganisms, 2023, doi:10.3390/microorganisms11020403_

Round 1

Reviewer 1 Report

Dear author

The manuscript "Inhibition of foodborne pathogenic bacteria by excreted metabolites of Serratia marcescens strains isolated from dairy producing environment" although provide some interesting information but still need significant changes and at several places the need more clearance in methods and improvements in results and discussion part.

I did not see any purified active compound in this study.

The antibacterial study of metabolites is usually carried out by agar well or agar disc method. But here, agar spot method was used which I think is not a more reliable test especially when other reliable tests are available.

It is good to add some pictures that are related to antibacterial activity like mostly the researchers add petri plates pictures that show inhibition zones, something like this. This will increase the quality of your work presentation.

The keywords should be different from title

The abstract seems totally descriptive, add some numerical digits for significant findings

What is TSA?

The source of pathogenic strains is not mention

In figures, instead of using different shapes, different colors are good option

What the bars indicate in figure 2?

Conclusion is a bit long having some useless description. Please concise and write in one paragraph

Reviewer 2 Report

The paper deals with the potential inhibitory activity of two strains of Serratia marcescens against some bacterial strains.

The study is interesting, but methodologic flaws are present. The prodigiosin detection is not convincing and no control was set up in the inhibition or co-culturing experiments. This impairs the entire experimental procedure, and without control, there is no possibility to ascertain the real significance if the results

Authors should repeat the experiments with the appropriate controls.

Data are poorly reported, most of them were neither been included in the manuscript, nor they are available in other repositories.

This impairs the Discussion, which is overall well-written but not supported by concrete data.

Following there are specific comments.

Line 32: Please use STEC in full as it is its first occurrence.

Lines 31-36: Please specify that those data are referred to EU.

Line 37: How is this point related to the previous?

Line 38: Why original? Please specify.

Lines 38-42: This statement report wide and complex aspects of microbial physiology, therefore the reference is not appropriate, as some specific studies, or, even better, reviews, should be used. Please avoid the excessive use of self-references, especially when not they do not fit with the content of the statements. Additionally, the sentence is identical to the one present in the cited study.

Lines 44-45: Please rewrite this sentence as there are syntax errors to be fixed.

Lines 45, 46, 48, and 49: Please italicize the species names.

Lines 50-53: Please check the punctuation of the sentence.

Line 90: Since it is the first occurrence, please write in full the specie name of Salmonella enterica (i.e. Salmonella enterica subsp. Enterica ser. Hartford).

Lines 94-97: This sentence is not clear. Another S. enterica strain was used, apart from S. Hartford NCAIM B.01310? Why is presented here? Please provide more details about this strain, at least the identification method and the antigenic formula.

Line 106-114: This sub-section should be merged with the previous one. To improve readability and comprehension of the procedure, I would also like to suggest providing more details in the text, other than citing the reference study.

Lines 99-114: No control without supernatant was set up.

Line 127: Checked? I suppose the diameter was measured. Please specify it in the text.

Line 135: Please provide more detail about the preparation of this medium.

Line 144: NH4 is a cation. Please add the positive charge to the formula or use NH3, if ammonia was used.

Lines 143-147: The procedure for determination and quantification of prodigiosin is not consistent with those reported in the cited references and it is not convincing.

Lines 150-152: Since they are cited in the Discussion section, primers should be cited here.

Line 152: Please specify that the gene chiA is related to the chitinase production.

Line 157: There is a number of methods, both biochemical and molecular, to identify S. enterica. Why 16S rDNA sequencing was used? Were there specific problems?

Lines 156-169: No control was used. Cultures with S. enterica and S. marcescens only should be contemporarily prepared for reference.

Lines 170-171. No detail is provided for those experiments.

Line 181: Please provide such data.

Lines 185-187: Please provide such data.

Lines 195-196: It should be advisable that experiments were carried out with all the strains, that would be used as a control.

Lines 196-199: Please check the syntax of this sentence.

Lines 213-214: Procedures should not be described in the figure legends but in the M&M section.

Lines 227-228: Without controls (i.e. growth without supernatant), it is not possible to establish if an actual lag does exist.

Lines 233-238 and 242-246: How this comparison was achieved if no control was prepared? The statistical tests were not detailed in the M&M section. Additionally, the results of the statistics should be presented in the manuscript. Please also provide the numerical data in appropriate tables to be included in the manuscript or made available as supplementary material.

Lines 246-248: This statement should be included in the Discussion section instead of the Results section.

Lines 250: No data about the clearing zones were produced.

Lines 239-241: Please check the manuscript structure.

Lines 267-268: This information should be included in the M&M section.

Lines 271-275: Considering that the amplicon dimension was not consistent with the expected, the nucleotide sequence determination should be performed.

Line 285: It is not clear why another strain of S. enterica was used. Was it associated with infection in humans or animals? Do the Authors consider S. Hartford somehow not suitable for this test?

Lines 288- 307: The significance of this data is questionable as no statistical analysis of data was applied. Control is only present in Figures, but no mention had been done in the M&M section.

Lines 292-292 and 302-303: How could the Authors assess that bacteria grow again? Were statistical tests performed? With the figure as the only visual data, random variations could not absolutely be excluded.

Lines 304-307: This statement is not appropriate for the Result section, it should be included in the Discussion section.

Lines 311-314: Please check this sentence.

Lines 315-323: A wide literature about S. marcescens as a contaminant exists.

Lines 350-356: In the light of those previous observations, the determination of the nucleotide sequence of all the amplicons is even more necessary.

Line 361: No affordable data has been produced to assess the significance of such differences.

Line 372: Please use “widely occurs” instead of “is widely occurred”.

Lines 375-377: Were those conditions ensured in the experimental tests?

Line 379: Please italicize “enterica”.

Lines 383-386: Analyses should be carried out to verify that diffusible proteins or other compounds might (r might not) affect the milk composition.

Lines 33-406: Please better address the Conclusion section, which should have the role of summing up all the discussed achievements.

Lines 397-402: The action of type VI secretion system has not been treated in the Discussion section.

Line 419: Please write “EFSA” instead of “EFSE”.

Reviewer 3 Report

Comments to the Author

The manuscript "Inhibition of foodborne pathogenic bacteria by excreted metabolites of Serratia marcescens strains isolated from dairy-producing environments" characterized the antibacterial activity of pigmented and non-pigmented S. marcescens strains isolated from dairy-producing environments and determined their potential applicability against bacterial pathogens in the food industry. The data presentation and methods used to generate the data are reasonable, but they should be clarified at some points. The authors should take the following points into consideration:

1.       Title” Inhibition of foodborne pathogenic bacteria by excreted metabolites of Serratia marcescens strains isolated from dairy-producing environments”

2.       line 17 “TSB”,  line 32 “STEC”, line 87 “TSA, TSB” & line 314 “RTE” please write in full at the first mention. 

3.       lines 62-65” Pigmented biotypes of S. marcescens could mostly be found in natural environments, whereas the non-pigmented biotypes are prevalent in the hospital, and non-pigmented strains are more frequent than pigmented ones among clinical isolates”: rewrite this sentence to avoid repetition

4.       lines 85-86: a diary-producing plant

5.       lines 84-86: Serratia marcescens subsp. marcescens CSM-RMT-1 and CSM-RMT-II-1 were identified using sequencing of 16S rDNA gene as mentioned in reference (23). GenBank accession No. of both strains should be mentioned.

6.       lines 87-88, 97: supplemented with 20% glycerol “delete of”

7.       lines 89-91: write the source of the tested strains

8.       lines 123-124: write the company and country of each component

9.       lines 139-140 “Nutrient agar plates”: write the company and country of medium instead of mentioning the ingredients.

10.   line 142” 96% ethanol

11.   All species should be italic through the manuscript

12.   lines 156-157: “Co-culturing study of a food-derived Sa. enterica strains (previously isolated by the authors from egg powder and identified by sequencing its 16S rDNA gene “:  S. enterica; write the GenBank accession No.

13.   line 160& 285& 379: S. enterica  & L 176 and in table 1& 203& 227: S. Hartford

14.   Adjust Figure 2 to be above the figure legend

15.   line 246: Nonetheless

16.   lines 273- 274 “Further optimisation regarding to the reaction compounds and the annealing temperature is needed for the detectability of this gene.” Did the authors obtain the predicted amplicons after optimization of conditions?

17.   line 286:TSB and milk

18.   Lines 304-306: avoid mentioning reference in the result section; it should be in the discussion instead.

19.   line 310: alimentary tract

20.   line 330 replaces “groups of eubacteria” with “bacteria”

Reviewer 4 Report

Bernadett Baráti-Deák et.al,  have submitted a manuscript entitled ‘Inhibition of foodborne pathogenic bacteria by excreted metabolites of Serratia marcescens strains isolated from dairy producing environment. The authors found that Serratia marcescens strains a dairy producing environment were tested for their inhibitory effect on  Listeria monocytogenes, Salmonella Hartford, Yersinia enterocolitica and Escherichia coli. Inhibition of foodborne pathogens was seen in case of a non-pigmented Serratia strain, while the pigment producing isolate was able to  inhibit only Y. enterocolitica. There co-culturing study in TSB and milk showed that growth of Salmonella was inhibited in the initial hours, but later the pathogen could grow in the presence of Serratia strain even if its cell concentration was 1000 times higher than that of Salmonella. However, authors found that, concentrated cell-free supernatants had stronger inhibitory activity, which confirms the extracellular nature of the antagonistic compound. Application of concentrated S. marcescens cell-free supernatant can be an effective antibacterial strategy in food industry, mainly in form of biodisinfectant on surfaces of food-processing areas.

Overall, the study is well organized and well-presented but before accepting for publication I have few suggestions.

Minor comments:

Result: 3.1: No original data was shown. At least 1 plate picture shows in the main figure and the rest of the picture is in the supplementary figure. How many times these experiments were performed, is nowhere mentioned in the text. Even I cannot see any SD values.

For figure 1A-d; How many biological replicates? No standard deviation in the graph. Can the author explain this?

Author Response

We had no reviewer 4.

Round 2

Reviewer 1 Report

Paper has been improved scientifically 

Author Response

We thank you again for your comments and suggestions to our original manuscript, all of which significantly contributed to increasing the scientific quality of the work.

Reviewer 2 Report

The manuscript has been definitely improved, and the major issue, specifically, the lack of control, has been fixed. However, minor concerns still remain, such as the sequence of the amplicons not corresponding to the expected size, and the omission of the statistic values.

I believe that those data can and should be provided, since Sanger sequencing of a small amplicon is a common and quite cheap procedure (about 10-20 $ for both strands), and there is no space problem for showing statistical data because this is not a printed journal.

Finally, the paper would benefit in clarity by the determination of the antigenic formula of the food-derived Salmonella enterica strain, that, obviously, cannot be achieved by MALDI-TOF or analysis of the 16S rRNA gene.

Minor linguistic problems should be fixed, such as, not exhaustively, those following listed:

Line 34: The genus name of a microbial species should be provided in full at its first occurrence.

Line 45: Please use “thus contributing” instead of “thus contribute”.

Line 54: It is not clear the meaning of “widespread antimicrobial resistance”. Is AR widely diffused? Where?

Lines 61-64: The verb is missing.

Reviewer 3 Report

The manuscript is revised according to the suggestions/corrections. Now it is suitable to be published in the journal. I have the following minor observation which the authors should address before the paper is considered for publication.

1-In Table 1: please write (bp) in the heading: Size of the PCR product (bp) and don’t write it in the rows. In the table, footnote write: bp: base pair

2-In Table 3: adjust (days of incubation)

Reviewer 4 Report

Thank you for adding the plate pictures.

Author Response

We had no Reviewer 4.